# Mitochondrial Bioenergy in Neurodegenerative Disease: Huntington and Parkinson

**DOI:** 10.3390/ijms24087221

**Published:** 2023-04-13

**Authors:** Annalisa Tassone, Maria Meringolo, Giulia Ponterio, Paola Bonsi, Tommaso Schirinzi, Giuseppina Martella

**Affiliations:** 1Laboratory of Neurophysiology and Plasticity, IRCCS Fondazione Santa Lucia, 00143 Rome, Italy; 2Saint Camillus International University of Health and Medical Sciences, 00131 Rome, Italy; 3Unit of Neurology, Department of Systems Medicine, Tor Vergata University of Rome, 00133 Rome, Italy

**Keywords:** movement disorders, mitochondria, energy metabolism, synaptic plasticity, basal ganglia, calcium, Parkinson’s disease, Huntington’s disease

## Abstract

Strong evidence suggests a correlation between degeneration and mitochondrial deficiency. Typical cases of degeneration can be observed in physiological phenomena (i.e., ageing) as well as in neurological neurodegenerative diseases and cancer. All these pathologies have the dyshomeostasis of mitochondrial bioenergy as a common denominator. Neurodegenerative diseases show bioenergetic imbalances in their pathogenesis or progression. Huntington’s chorea and Parkinson’s disease are both neurodegenerative diseases, but while Huntington’s disease is genetic and progressive with early manifestation and severe penetrance, Parkinson’s disease is a pathology with multifactorial aspects. Indeed, there are different types of Parkinson/Parkinsonism. Many forms are early-onset diseases linked to gene mutations, while others could be idiopathic, appear in young adults, or be post-injury senescence conditions. Although Huntington’s is defined as a hyperkinetic disorder, Parkinson’s is a hypokinetic disorder. However, they both share a lot of similarities, such as neuronal excitability, the loss of striatal function, psychiatric comorbidity, etc. In this review, we will describe the start and development of both diseases in relation to mitochondrial dysfunction. These dysfunctions act on energy metabolism and reduce the vitality of neurons in many different brain areas.

## 1. Introduction

The mitochondrion is an evolutionary organelle that originated from the established symbiotic relationship between alpha-proteobacteria and eukaryotic cells [1]. This organelle was stabilized in the host through mitotic cell divisions [1,2]. In light of this hypothesis, it is possible that the mitochondria enhanced the development of life as we know it and represent the common factor through which many organisms function or dysfunction [2,3,4]. Mitochondria are well-known energetic units of eukaryotic cells [2]. They supply energy in the form of ATP to support all cellular functions. In aerobic respiration, through a complex biochemical pathway that involves the addition of different co-factors, the mitochondria aid in the synthesis of ATP molecules from pyruvate [2,3]. In addition, mitochondrial proteomes, which are aggregates of more than 1000 proteins, can support a wide variety of critical biochemical processes, including amino acid metabolism, nucleotide metabolism, protein synthesis, and fatty acid catabolism [3,4].

There is abundant evidence in the literature associating mitochondrial disruption with neurological and neurodegenerative diseases [4,5,6]. The brain requires high-energy consumption in the form of ATP. In fact, it was estimated that a cortical neuron in the human brain could utilize 4.7 billion molecules of ATP/sec. Furthermore, the ATP used at rest corresponds to about 5.7 kg per day [7]. The brain’s high energy demand supports many important functions, such as neurotransmission, cellular activities required for learning and memory, neural plasticity, and synapse development [8]. Since the mitochondrion, through oxidative phosphorylation and the production of ATP, meets all the energy demands of the nervous system, even with a rapid turnover [9], mitochondrial dysfunction with a consequent loss in the energetic capacity may represent a neurodegenerative trigger. Severe bioenergetic dysfunction makes neurons very vulnerable to oxidative stress and predisposed to neuronal cell death [10]. Additionally, neurodegeneration can be seen as an energy disorder, which manifests in different aspects depending on the severity of the process and the area involved [11]. This energetic disorder can lead to different pathologies, but sometimes with opposite symptoms, such as hypokinetic and hyperkinetic disorders. The purpose of this review is to emphasize how mitochondrial dysfunctions, which may be associated with phenomena such as inflammation and degeneration, are actually a common element in two major neurodegenerative diseases. One is purely caused by genetics, while the other results from a combination of genetic and environmental factors: Huntington’s disease (HD) and Parkinson’s disease (PD), respectively.

## 2. Role of Mitochondria in Brain Energy Metabolism, Calcium Homeostasis, and Signal Transduction

Mitochondria are organelles responsible for several critical processes in neuronal function and dysfunction, including energy metabolism, calcium homeostasis, and signal transduction [1]. The brain consumes about 20% of the total oxygen respired to meet the high metabolic demand of neurons that must maintain ionic gradients across membranes, transport molecules from the soma along axons and dendrites, as well as for neurotransmission [12]. The metabolic activities that generate ATP rely mainly on mitochondrial oxidative phosphorylation (OXPHOS) [1]. During the process of OXPHOS, enzymes of the Krebs cycle utilize acetyl-coenzyme A to reduce the cofactors, nicotinamide adenine dinucleotides (NADH) and flavin adenine dinucleotides (FADH_2_), that aid energy transfer to the electron transport chain (ETC), embedded within the extensive inner mitochondrial membrane [13,14]. The ETC consists of four complexes that transfer electrons from NADH and FADH_2_ to O_2_. Ultimately, the energy released by the transfer of electrons is used for the thermodynamically unfavorable pumping of protons against their concentration gradient from the matrix to the intermembrane space to generate an electrochemical gradient known as mitochondrial membrane potential (ΔΨm) [1]. This membrane potential is essential for the process of energy storage (Figure 1). The protons accumulated in the intermembrane space are then allowed to move according to a concentration gradient back into the matrix by passing through one of the domains of the enzyme ATP synthase (or Complex V), which finally harnesses the released energy to phosphorylate ADP molecules into ATP [13]. Proper neuronal function is strongly linked to the retention of mitochondrial membrane potential and ATP levels [15] (Figure 1). 

Indeed, mitochondrial functions go beyond the primary role of ATP production because these organelles are intimately involved in numerous processes operating within the cell, including calcium homeostasis, the generation of free radical species (ROS), steroid synthesis, apoptosis, and cell signaling pathways [16]. Particularly, the direction of the mitochondrial membrane potential (negative internal) has important implications, as it elicits the thermodynamic force supporting the accumulation of metal cations, especially calcium (Ca^2+^) in the mitochondria [17]. Mitochondrial Ca^2+^ homeostasis plays a key role in cellular bioenergetics and signaling [18]. Ca^2+^ passage across the outer mitochondrial membrane (OMM) is mediated by the VDAC channel, whose different selectivity for cations or anions is voltage-dependent [19]. For instance, at low potentials (10 mV), the VDAC channel is highly permeable to anions and able to maintain a low Ca^2+^ flux. Conversely, increases in the membrane potential (20–30 mV) result in a conformational change that allows a 4- to 10-fold increase in Ca^2+^ influx [20]. In comparison, the inner mitochondrial membrane (IMM) has considerable Ca^2+^ permeability: the mitochondrial calcium uniporter (MCU) contributes to the potential-dependent Ca^2+^ influx into the mitochondrial matrix, while the mitochondrial Na^+^/Ca^2+^ exchanger (NCLX) is one of the main units involved in Ca^2+^ extrusion [21]. In general, cytosolic calcium uptake by mitochondria will occur only if they are exposed to elevated Ca^2+^ concentrations [22,23] (Figure 1). However, within the cell, Ca^2+^ is compartmentalized, and the average resting cytosolic concentration is remarkably low [24,25,26]. One of the main stores of cellular calcium is the endoplasmic reticulum (ER), which contributes to orchestrating cellular Ca^2+^ homeostasis through its interaction with the mitochondria [27]. Notably, Ca^2+^ uptake in the mitochondria probably occurs only at sites in close proximity between the ER and mitochondria [18]. On the other hand, neuronal mitochondria, unlike those in non-neuronal cells, have a considerably lower threshold for Ca^2+^ uptake. In addition, ER-mitochondria contacts are critical for Ca^2+^ uptake in dendritic mitochondria but not in axonal mitochondria [28]. This property of axonal mitochondria is due to the presence of the brain-specific uniporter MICU3, which causes axons to be less dependent on intracellular Ca^2+^ storage. In the absence of MICU3, synaptic function is impaired [29] (Figure 1).

Mitochondrial calcium-signaling in neurons regulates metabolism and energy production, which are crucial for neurotransmission and sustaining synaptic plasticity [30]. In addition, mitochondrial Ca^2+^ also drives the production of reactive oxygen species (ROS) [31]. Mitochondria are the primary sources of ROS in cells and actively participate in cellular redox regulation and ROS signaling [32] (Figure 1). ROS, in the form of superoxide, are natural byproducts of normal mitochondrial activity and are naturally converted to H_2_O_2_, which is in turn scavenged by the enzyme catalase to produce water [32]. Under normal conditions, through the generation of ROS and redox signaling, mitochondria can control cell metabolism and physiology, as well as inflammatory responses, immune function, autophagy, and stress responses [33,34,35]. Indeed, ROS, and particularly hydrogen peroxide at low concentrations, act as important signaling molecules in the cell, activating several protein kinases, such as PKA, PKC, PI3K, and p38 [36]. Moreover, in immune cells, mitochondrial metabolites and ROS finely regulate signaling pathways and cell fate, thereby orchestrating the immune response [37].

The immune system comprises a diverse family of cells with multiple roles during homeostasis and inflammation, capable of using distinct metabolic programs to undertake their functions. For instance, during the immune response, effector T cells promote aerobic glycolysis, while memory T cells and regulatory T cells promote fatty acid oxidation [38]. Particularly, the mitochondria can modulate the metabolic and physiological states of different types of immune cells [39]. It can also stimulate the innate immune signaling cascade, which can intensify inflammation following cytotoxic stimuli or microbial infection [40]. For example, the innate immune receptor NLRX1, a member of the Nod-like Receptor (NLR) family, is located in the mitochondria and undertakes an important role in maintaining cellular homeostasis following acute mitochondrial injury [41,42]. In addition, it has recently been discovered that mitochondria play a central role in initiating and regulating the NLRP3 (nucleotide-binding domain, leucine-rich repeat family, pyrin domain-containing 3) inflammasome [43]. This multiprotein complex, activated upon infection or cellular stress, leads to the secretion of proinflammatory cytokines, such as interleukin-1β (IL-1β) and IL-18, that trigger an inflammatory form of cell death called pyroptosis [44,45]. Although the mechanisms of NLRP3 inflammasome activation are still debated, it is widely believed that changes in the mitochondrial membrane potential, permeabilization of the outer mitochondrial membrane, and increased formation of mitochondrial ROS are crucial factors in inducing the cytosolic translocation of mitochondrial molecules such as cardiolipin and mitochondrial DNA, which are capable of activating the inflammasome [43]. Indeed, the overproduction of ROS and dysregulation of the redox signaling system result in oxidative stress that can lead to mitochondrial damage, induce mitochondrial DNA mutations, damage the respiratory chain, alter membrane permeability, and affect the Ca^2+^ homeostasis and mitochondrial defense systems [46]. Accumulating evidence suggests that oxidative stress and mitochondrial injury can result in cellular DNA damage, degradation of proteins and lipids, and the pathogenesis of neurodegenerative diseases [47,48]. Nevertheless, cells have an accurate endogenous antioxidant defense system that can maintain cellular redox homeostasis between ROS production and elimination to ensure normal cellular signaling and redox regulation (Figure 1) [49].

The nuclear factor erythroid 2–related factor 2 (Nrf2) is an emerging therapeutic target, since it is involved in cellular resistance to oxidants. In detail, Nrf2 controls the basal expression of genes coding for enzymes and proteins involved in antioxidant and detoxifying action, repair and removal of damaged proteins and organelles, the inflammatory response, and mitochondrial bioenergetics [50,51]. Additionally, malfunctioning mitochondria can be selectively removed through a conserved cellular recycling process known as mitochondrial autophagy or mitophagy. In fact, the efficient elimination of damaged mitochondria prevents activation of cell death pathways, protects against ROS overproduction, and maintains efficient ATP production [52]. Damaged mitochondria are swallowed into autophagic vesicles that subsequently transport them to lysosomes for destruction. Mitophagy is a strictly regulated process, modulated by mitochondrial fission and fusion proteins, BCL-2 (B-cell lymphoma 2) family proteins [53], and the PINK1/Parkin pathway [54]. As all other defense mechanisms fail, the neuron can orchestrate its own-destruction by activating an intrinsic suicide program, otherwise known as apoptosis [55] (Figure 1). The underlying mechanisms of apoptosis are very sophisticated and engage an energy-dependent cascade of molecular events. In addition, it can follow different molecular pathways, one of which is the intrinsic pathway that involves the mitochondria [56]. Indeed, these organelles represent the site where anti-apoptotic and pro-apoptotic proteins interact and the origin of signals that initiate the activation of caspases, the cysteine proteases capable of cleaving many cellular substrates to disrupt cellular contents [57].

It is therefore understood that mitochondrial integrity and homeostasis are prerequisites for proper cellular functioning, especially for neurons, which are polarized, complex cells with high energy demands [58]. Not surprisingly, neurons have the highest content of mitochondria compared to other cell populations. Mitochondrial functionality is essential for ensuring membrane excitability and performing the complex neurotransmission and plasticity processes.

Firstly, the mitochondria provide the energy needed to undertake a wide range of neuronal functions, such as the maintenance of resting membrane potential, the restoration of ionic balance after depolarization, the cycling of synaptic vesicles, and the transport of proteins and organelles from the soma to distal sites. Interestingly, the mitochondria in axons and dendrites have different morphologies, some are small and sparsely distributed in the former, whereas others are elongated and densely distributed in the latter. In addition, axonal and dendritic mitochondria differ in movement, metabolism, and responses to neuronal activity [59].

Secondly, several lines of evidence support the role of mitochondria in the mobilization and recycling of synaptic vesicles [60]. Neurotransmission is underpinned by endocytosis and the local filling of synaptic vesicles in the presynaptic terminal [61]. Furthermore, the mitochondria support synaptic activity through cytosolic calcium reabsorption, a critical buffering mechanism for establishing and maintaining synaptic activity and preventing neuronal toxicity and excitotoxicity. Moreover, mitochondrial calcium buffering appears to be necessary at the synapse, even when the neuron is at rest: synaptic terminals lacking mitochondria show a higher frequency of spontaneous release of neurotransmitter-containing vesicles [62].

Overall, the importance of the mitochondria in neurons is unequivocal. Therefore, mitochondrial dysfunction leads to a plethora of severe conditions, from impaired neuronal development to various neurodegenerative diseases.

## 3. Role of Mitochondria in Neurodegenerative Disease

A great deal of evidence shows the effects of mitochondrial dysfunction on degenerative diseases [63]. Diseases caused by mitochondrial alterations often show a neurodegenerative component involving the nervous system. Similarly, mitochondrial defects are frequently observed in tissue samples taken from patients with a neurodegenerative disorder. This evidence reflected the high-energy requirements for all biological processes [64]. The continuous increase in energy demand, mainly due to excess food consumed daily, the constant increase in energy required for thermoregulation, or the increase in energy needed to counteract the environmental toxicity caused by humans, leads to an overwork of the mitochondria, resulting in impaired bioenergy efficiency [64,65]. It was demonstrated that cells exposed to an opulent nutrient environment are inclined to have their mitochondria in a fragmented state. Moreover, the mitochondria observed in cells in malnourished conditions remain for a longer period of time in the associated state [66,67]. This portrays the fact that the mitochondria can change their architecture and, therefore, their bioenergy capacity according to external events.

A degenerative disease is regarded as a type of non-physiological condition whose origins in a tissue or organ worsen over time. Energy failure has been linked to degenerative processes such as cancer, aging, neuroendocrine disease, neurodegenerative disease, and inflammatory diseases. Likewise, results from many studies support the involvement of the mitochondria in neurodegenerative diseases, particularly PD [68,69,70,71,72]. In addition, studies relating to the MPTP toxin first highlighted the role of mitochondrial complex I dysfunction and neurodegeneration in PD [73]. In association with other commonly used toxins, such as rotenone and paraquat, MPTP offered a new insight into defining the effective role of mitochondrial bioenergy in neurodegeneration [74]. Indeed, more than five thousand manuscripts dealing with the association between Parkinson’s disease and mitochondrial alteration can be found on PubMed (Figure 2). In addition, due to studies on PD, particularly genetic Parkinsonism, all mitochondrial dysfunctions leading to cell death have been well defined [75]. The PINK1 gene mutation, responsible for an early onset of Parkinsonism, serves as a good example [76]. This gene codes for the mitochondrial protein, phosphatase, and tensin homolog serine/threonine-protein kinase 1 (PTEN-induced kinase 1) [77]. PTEN-proteins can protect cells against oxidative stress, proton chain dysfunction, and bioenergy failure [76,78]. The PINK1 gene has also been identified as an oncogene with tumor suppressor properties [77,79,80]. Additionally, the PINK1 protective role has been observed in many disorders characterized by progressive inflammation and neurodegeneration, such as Alzheimer’s disease, multiple sclerosis, amyotrophic lateral sclerosis, and HD [77]. In physiological conditions, PINK1 is translocated inside the mitochondria in its mature isoform with the aim to withstand the activity of the mitochondrial chain and to produce (at the level of complex I) the molecules of ATP showing the maximal bioenergetic efficiency (Figure 1). Moreover, PINK1 is also located in the inner mitochondrial membrane and can interact with the chaperone TRAP1 (also recognized as an interactor of the type 1 tumor necrosis factor receptor) to maintain important bioenergetics and proteostatic functions [77,81]. Cleaved PINK1 can interact with other chaperon proteins, and when it is located in the cytosol, it can activate the m-Tork/Atk pathway. In addition, PINK1 can mediate the phosphorylation of another important gene for PD: Parkin. The coupled action of PINK1 and Parkin may induce mitochondrial fusion in order to cause the elimination of dysfunctional mitochondria. Furthermore, PINK1 is also involved in the formation of an autophagosome complex by the activation of beclin protein and may regulate the apoptotic process [77]. The role of the PINK1 gene in PD was well investigated by the use of animal models and cellular cultures of human fibroblasts [82,83,84,85,86,87] (Table 1). These studies have confirmed the role of bioenergetic efficiency in the maintenance of a healthy state or the progression of neurodegeneration. However, the absence of PINK1 results in low bioenergetic efficiency and programmed cell death after each minor stress [86,88]. It is very interesting to note that PINK1 not only has a key role in degeneration, oncological degenerative processes, and neurodegenerative PD but also in the neurodegeneration found in HD. Parkinson’s disease (PD) and HD are two different neurological diseases involving the central nervous system. The symptomatology of both is quite similar (cognitive impairment, limb inflexibility, and problems walking or talking), but while PD results from a combination of genetic and environmental factors, HD is only an inherited genetic disease. Both pathologies have a common factor, which is the loss of bioenergetic efficiency [77,89].

Huntington disease is an incurable degenerative disorder caused by a mutation in the huntingtin gene, where the CAG sequence is excessively repeated. This mutation alters numerous cellular processes and leads to cell apoptosis. One important alteration is caused by the impairment of the mitochondrial metabolism. In a study conducted on the fly model, the formation of an abnormal ring-shaped mitochondria was observed; this particular shape was previously identified in mitophagy-blocked cells in which PINK1 overexpression was able to rescue the regular shape and function of the mitochondria. PINK1 over-expression was able to improve bioenergetic efficiency (increasing ATP levels) and rescue neuronal integrity in the adult drosophila model of HD [89]. HD is a degenerative pathology caused by the pathological expansion of CAG repeats in the Huntington gene, which codes for the Huntingtin protein. The gene is located on chromosome number 4 and is characterized by high levels of polymorphism. Unlike PD, HD has an age range of about 35–44 years and an estimated post-onset life span of 15/18 years. It is also characterized by an advanced motor disability, including chorea [111]. Enough evidence suggests that the mutation may cause an alteration in mitochondrial trafficking, an increase in oxidative stress, dyshomeostasis in intracellular calcium content, an alteration in bioenergetics, and an alteration in a respiratory mitochondrial chain [112,113,114,115]. Huntington’s disease, PD, and Alzheimer’s disease are three neurodegenerative diseases that have 37 common genes and about 40% of whose products act at the mitochondrial level [116]. These neurodegenerative diseases are coupled to a physiological degenerative process called aging or senescence that starts at the mitochondrial level and results in reduced bioenergetic efficiency [117]. Cellular senescence is characterized by heavy changes in cellular metabolism and an increase in pyruvate utilization that may produce differences in phosphorylation states, thereby increasing the activity of the mitochondrial pyruvate dehydrogenase complex and ROS production [118,119]. These features may lead to an impairment of cellular function that induces degeneration, as in the case of cancer or other neurodegenerative processes [120]. Moreover, recent data have demonstrated that key oncogenes and tumor suppressors modulate mitochondrial metabolism and dynamics. Indeed, different types of cancer result in more or less sensitivity to the modulation of mitochondrial function in the lesion caused by the tumor [121]. Therefore, the mitochondria play a cardinal role in many degenerative processes, which are mostly demonstrated in animal models.

## 4. Mitochondria Bioenergy in Parkinson’s Disease and Huntington Disease in Rodents Animal Models

Parkinson’s and Huntington’s diseases are both neurodegenerative diseases with multiple mitochondrial bioenergy alterations related to metabolism, oxidative stress, dynamics of biogenesis transport, and mitophagy (Table 1). As a result of the close relationship between neurodegeneration and mitochondrial bioenergy, one might presume that changes in mitochondrial homeostasis with the release of calcium and NO, as well as reactive oxygen species (ROS), will affect the unfolded protein response, modulating specific signaling molecules, and reprogramming mitochondrial bioenergy [122] (Table 1). The mouse models used in the study of PD and HD serve as valuable tools in defining the role of mitochondrial bioenergy in the pathogenic mechanisms of these diseases (Table 1).

### 4.1. Parkinson Disease

#### 4.1.1. Neurotoxin-Induced and Autosomal-Dominant PD Models

Parkinson’s disease animal models are associated with multiple mitochondrial defects for review [122,123,124,125] (Table 1). Indeed, human idiopathic PD patients showed abnormalities in mitochondrial activity with reductions of complex I, NADH: ubiquinone oxidoreductase, and strikingly reduced succinate: cytochrome c oxidoreductase, suggesting an etiological role in the pathogenesis of PD [126].

Chemically, the neurotoxin agent is widely used to recapitulate Parkinsonian features in various animal models [127,128]. In fact, the neurotoxins 1-metil-4-fenil-1,2,3,6-tetraidropiridina (MPTP), rotenone, and paraquat block mitochondrial bioenergetics in dopaminergic neurons and induce Parkinsonian syndrome for review [129,130,131,132,133,134,135]. However, it is important to consider that overexpression of α-synuclein or agents that generate stress or cytosolic acidification show the translocation of α-synuclein into the mitochondria or in the mitochondrial membrane [136,137,138]. α-synuclein is a presynaptic neuronal protein that is linked to familial and idiopathic PD [139]. Both the over-expression and loss-of-function of the SNCA gene contribute to the manifestation of the disease [140], according to the “α-synuclein cascade hypothesis” [141]. 

Experimental studies in A53T mutant mice, a transgenic animal model that expresses human α-synuclein, show that they develop mitochondrial DNA damage and degeneration with an apoptotic-like death of neocortical, brainstem, and motor neurons [91]. In the same animal model, mitochondrial complex IV activity was reduced significantly in the spinal cord [142]. Additionally, in vivo human α-synuclein expression is associated with a decrease of Drp1 (Dynamin-related protein 1), a major player in the regulation of mitochondrial dynamics and the maintenance of their proper function [91]. On the other hand, the dysfunction of Drp1 in the mitochondria is also associated with enlarged neuronal mitochondria [91]. Other proteins, such as Mitofusin1 (Mfn1) [91], that facilitate mitochondrial fusion are decreased in these models, which correlate with the mitochondrial bioenergy dynamic changes. Previous studies in A53T mice suggest that α-synuclein distresses the mitochondrial morphology and reduces both Mfn1 and Mfn2 in an age-dependent manner [90]. In addition, an in vivo study in midbrain dopaminergic neurons suggested that α-synuclein predominantly accumulates in the central mitochondrial membrane and interacts with complex I, resulting in impaired activity of the mitochondrial electron transport chain [92]. Consistent with these findings, other experimental analyses show that after post-inoculation of human α-synuclein in rats, its accumulation in striatal dopaminergic terminals from the SNpc and early synapse loss were observed [143]. To explain this early synapse loss, a proteomic analysis using a real-time cell metabolic method and isolated synaptosomes was used to investigate the functional and molecular mechanisms. Upon injection in isolated striatal synapses, multiple dysfunctions of mitochondrial bioenergetics and morphology were detected with the upregulation of PRKAG2 and TTR, respectively, which are sensor proteins of the cell energy status and markers of oxidative stress [143]. Moreover, ultrastructural examination of subsequent human α-synuclein accumulation showed an altered expression of Rab5 endocytic and LC3 autophagic proteins that potentially reflect an accumulation of autophagic vesicles [143]. This study allowed the evaluation of an interesting aspect, namely the effect of oxidative stress and the mitochondrial bioenergy alteration on both dopaminergic neuron death and striatal neurons. These could be both responsible for the induction of Parkinsonian syndrome.

The neurons are high-energy consumers and employ most of the energy at the level of the synapse to preserve and restore ionic gradients and for the uptake and recycling of neurotransmitters [93]. Various mouse lines expressing a different type of α-synuclein present impaired dopamine neurotransmission and striatal synaptic plasticity [144]. In rodents, the injection of α-synuclein with an adeno-associated viral vector blocks the induction of long-term potentiation (LTP) and long-term depression (LTD) that are normally expressed in medium spine neurons (SPNs) [135,145,146,147], which are associated with early memory and motor alterations. There is sufficient evidence showing that α-synuclein misfolding and aggregation lead to mitochondrial stress, which appears to be related to the dysfunction of synaptic plasticity. Another interesting aspect is that α-synuclein overexpression via the endoplasmic reticulum (ER), promotes Ca^2+^ transfer in the mitochondria, while its silencing impairs mitochondrial function by loosening the ER-mitochondria interface [148]. However, the precise mechanism by which α-synuclein accomplished pathological characteristics and specific cell death remains obscure. Nevertheless, it is important to note that α-synuclein plays a pro-apoptotic role in different neuronal cells [149]. The wild-type α-synuclein can defend neurons from apoptosis by inhibiting caspase-3, after which the mutant α-synuclein loses this activity [149]. Caspases-3 actions then cause general damage and degeneration, aggregate in synapses, and persist in neurons without causing acute cell death [150,151,152] (Table 1).

#### 4.1.2. Autosomal-Recessive PD Models

The discovery of many genes related directly or indirectly to the mitochondria in PD underlines the implication of mitochondrial bioenergy in its pathogenesis. For instance, genes linked to early-onset recessive PD include: Parkin [83], PINK1 [76], and DJ-1 [153]. These genes have been strongly implicated in the mitochondrial morphology, damage, and degradation via mitophagy [154], which is involved in the mitochondrial activity and quality control mechanisms [77,155,156,157]. Parkin encodes a protein localized in the cytoplasm that contains an *N*-terminal ubiquitin-like domain with a function as E3 ubiquitin-protein ligase [158,159,160]. Parkin knockout (KO) mice show alteration in the levels of different proteins involved in detoxification, stress-related chaperones, without neuronal degeneration [161]. Parkin also has neuroprotective effects through the regulation of different cellular processes or pathways, such as mitochondrial swelling and cytochrome c release [100,162,163]. Normally, Parkin resides in the cytoplasm but can translocate to depolarized or damaged mitochondria to mediate their removal by mitophagy in cooperation with PINK1 and potentially other factors [164]. In a mouse model, Parkin KO was reported to cause a decrease in subunits of complexes I and IV with a reduction in peroxide reductases. In addition, the serum also aids in the reduction in antioxidant capacity and increases protein levels of lipid peroxidation [165]. Other studies on Parkin-deficient mice showed an increase in the content of dopamine in the striatum and the consequent reduction in the excitability of striatal SPNs [101]. Deficits in glutamate neurotransmission and amphetamine-induced dopamine release were also observed in other Parkin mutant mice with an increase in the metabolism of dopamine (MAO) [166]. Functional analysis of striatal SPNs showed an impairment of bi-directional corticostriatal synaptic plasticity with the loss of LTD and LTP in Parkin KO mice, while synaptic plasticity was not altered in the hippocampus of these animals [167]. The dopaminergic defect [101,166] may explain the selectively enhanced sensitivity to the striatal group II metabotropic glutamate receptor in cortically evoked excitatory post-synaptic potentials recorded from SPNs. This reinforcing effect is an adaptive change [78]. Furthermore, recent evidence in Parkin-KO rats suggests that the lack of Parkin is also necessary for the maintenance of post-synaptic endocytosis of AMPARs and for the decreased expression of the post-synaptic protein Homer1, which is essential for coupling the AMPA receptor endocytic zones with the post-synaptic density [168]. On the other hand, changes to the numbers or function of ER-mitochondrial contact sites may also affect mitochondrial calcium homeostasis, which is regulated by Parkin [169]. Calcium uptake into the mitochondrial matrix results in enhanced respiratory function, thereby tuning synaptic activity [60,170]. Recent studies also showed that the conditional Parkin KO in adult animals expresses the progressive loss of dopamine neurons. Moreover, the overexpression of PGC-1α leads to the selective loss of DA neurons in the substantia nigra [171].

Another gene linked to early-onset recessive PD is PINK1, which encodes a protein with a serine/threonine kinase catalytic domain, whether cytosolic or mitochondrial-associated [77]. PINK KO mice models exhibited significant functional impairment of synaptic plasticity and mitochondrial morphology without motor deficits or dopaminergic neuronal loss [83]. Consistent with these findings, PINK KO mice models generated with the silencing of the PINK1 gene did not develop dopaminergic neurodegeneration [172]. The PINK1 heterozygous KO mouse also showed selective impairment of LTP with a normal expression of LTD [173]. Nevertheless, low doses of rotenone were sufficient to induce severe alterations of the corticostriatal LTP and LTD [84]. This led to the proposal that the PINK1 homozygous KO model represents a good model to study the effect of gene-environment interaction. Although PINK1 KO rats show mitochondrial alteration, locomotor deficits, and α-synuclein aggregate in several brain regions, such as the cerebral cortex and dorsal striatum, it is also responsible for the degeneration of the substantia nigra [174,175,176,177]. However, this model presents clear symptomatic deficits that occur between 4 months and 9 months [176]. In the same model, using magnetic resonance spectroscopy, the mitochondrial metabolomic alterations increased in the cortex and striatum during the asymptomatic period, which coincides with the normal metabolic alterations [176]. Functional and bioenergetics experiments revealed that mitochondrial alterations in PINK1 KO rats are due to impaired complex I respiration, a dysfunction in ATP synthase, and reduced substrate oxidation [175]. New scientific research performed in animal models with the deletion of PINK1 mice provided an indication that a lack of PINK1 results in impaired synaptic plasticity, which is caspase-mediated [86]. Caspase-3 presents an additional role in neuronal processes, including the regulation of synaptic function and pruning [95,151,152]. Previous studies of PINK1 KO show a loss of LTP and LTD with alterations in dopamine release [83]. Indeed, low activation of caspase-3 is critical in restoring cortical-striatal LTD [86]. However, high activation is involved in degenerative processes [152]. In PINK1 KO mice, analysis with electron microscopy and other functional studies of the striatum revealed significantly enlarged mitochondria and impaired activity [178]. Likewise, the functional examination showed impaired mitochondrial respiration and aconitase activity in the striatum but not in the cerebral cortex [178]. Additional studies of PINK1 KO mice showed a decreased dopamine release in the dorsal striatum in an age-dependent manner with an impairment of the basal mitochondrial respiration [88].

DJ-1 gene mutations cause autosomal recessive early-onset PD. DJ1 encodes a protein of the superfamily that belongs to the ThiJ/PfpI-like. Under normal conditions, DJ-1 is confined throughout the cytoplasm of neurons, with a small amount localized to the mitochondrial matrix and intermembrane space [179], indicating a role in the balance of the mitochondrial physiology [17]. In a study, it was reported that DJ-1 KO mice failed to express PD features such as substantia nigra degeneration or the formation of protein inclusions similar to PINK1 and Parkin KO mice [180]. Recent experiments in DJ-1 KO mice showed increased calcium influx into the neuron during its activity, creating basal mitochondrial oxidant stress [94,181]. Functional analysis of isolated mitochondria showed increased ROS and decreased aconitase activity in DJ-1 KO mice [182]. Additionally, DJ-1 gene deletion revealed a deficit in scavenging because the loss of the DJ-1 pathway resulted in the loss of atypical peroxiredoxin-like activity [183]. In contrast, an independent study of DJ-1 KO mice reported an increase in mitochondrial respiration-dependent H_2_O_2_ consumption, mitochondrial Trx activity, total glutathione (GSH and GSSG, respectively) levels, mitochondrial glutaredoxin (GRX) activity, and a decrease in mitochondrial glutathione reductase (GR) activity [96]. Furthermore, new research on the semi-quantitative measurement of cerebral metabolites in DJ-1 KO mice showed significantly increased glutathione (GSH) levels and GSH/glutamate (Glu) ratio in the prefrontal cortex [97] (Table 1). It is important to note that the GSH system is more important than the antioxidant system in ROS detoxification in the brain [98,99,184]. Accumulated scientific information hypothesized that the increased sensitivity to oxidative stress in DJ-1 KO mice and dopaminergic neuronal cells is related to a decrease in ROS scavenging arising from deteriorated peroxidase-like scavenging with a deficiency of Nrf2 transcriptional factors and increased mitochondrial dysfunction due to a complex I deficiency [182,185,186,187,188]. Although DJ-1 KO mice had normal corticostriatal LTP, LTD was absent [101]. Overall, the data from animal models provide a framework that suggests that genes linked to early-onset recessive PD are important for maintaining normal mitochondrial bioenergy function (Figure 3). Gradually, the mutation protein described above, which is related to mitochondrial bioenergy, activates the response to mitigate mitochondrial alterations. Indeed, the genetic inactivation of PINK1/Parkin/DJ-1 with a triple KO mouse exhibits abnormalities in the mitochondrial pathway, activity, or morphology without PD-related phenotypes [189].

### 4.2. Huntington Disease

With regards to HD, several transgenic mice have been engineered to address specific pathological characteristics [190]. Multiple scientific information sources implicate that the gain and/or loss of function of mutant HTT leads to impaired mitochondrial bioenergy with alteration of oxidative stress, dysfunction of mitochondrial trafficking, and mitochondrial calcium dyshomeostasis [115]. In fact, malonate and 3-nitropropionic acid (3-NPA) are considered potent neurotoxins and are used to induce experimental HD in rodents using pharmacological models [191]. These oldest models provided the first piece of evidence that mitochondrial bioenergy, targeting the electron transport chain, is involved in the pathophysiology of HD. However, degenerating mitochondria have been identified in different areas, including the brain, liver, and muscle, in genetic mouse models of HD [192,193,194] (Table 1).

Both R6/1 and R6/2 strains express a single copy of a human genomic fragment that contains 116 and 144 CAG repeats of HTT under the control of the human HTT promoter [195]. These mice express a relatively rapid onset and progression of symptoms that include motor defects and neurodegeneration. Another major contribution to the research was the generation of knock-in mice models where CAG was repeated extensively (CAG94 and CAG140), compared to the human trans gene, for developing HD-associated phenotypes [190]. The R6/2 mice models have been the most widely used because the disease starts earlier and progresses more rapidly compared to the YAC, BAC, and other knock-in lines [196]. Transgenic animal models expressing the amino (N)-terminal fragment of the mutant form of HTT (R6/2) show an increase in 8-hydroxy-2-deoxyguanosine OH(8)dG during the late stages of the illness [197]. Therefore, oxidative stress occurs in the striatum before the onset of motor symptoms. In addition, biochemical analysis performed in the striatum of R6/2 mice at 12 weeks showed a significant reduction in the mitochondrial complex IV activities and a decrease in aconitase in the cerebral cortex [198]. In accordance with these data, new scientific research using synaptosomes isolated from R6/2 mice showed a decrease in the mitochondrial mass with increased ROS production and antioxidant levels in the striatum compared to the cortex [199], indicating oxidative stress. Additionally, in the same model in depolarized conditions, the oxygen consumption rates in synaptosomal cells were significantly amplified, which was accompanied by a clear increase in the mitochondrial proton leak of the striatal synaptosomes [199], indicating synaptic mitochondrial stress. Indeed, synaptic dysfunction occurs in various mouse models with HD, which appears to be associated with the dysfunction of LTD but displays a physiological LTP with lost synaptic depotentiation. These have been directly connected with the onset of cognitive deficits in HD models [103,107,108,200,201]. Using the brain slices from R6/1 mice at different ages, we observed a significant increase in Ca^2+^ content after glutamate stimulation. Such alterations have an impact on mitochondrial dysfunction with a decrease in NAD(P)H fluorescence and a loss in mitochondrial membrane potential (ΔΨm) [202]. Several groups characterized the reduction in *N*-acetyl aspartate in R6/2 mice, similar to symptomatic HD patients [102,203,204,205,206], and noticed fluctuations in some metabolites, which suggest the impairment in cellular metabolism and mitochondrial bioenergy [207]. 

Another interesting contribution to assessing the deficits of mitochondrial function and redox deregulation is the experimental data performed with PET in YAC128 transgenic mice [104]. PET analysis displays a specific accumulation of [64Cu]-ATSM in the striatum with concomitant alterations in mitochondrial respiration and ATP production associated with increased complex II and III activities. In turn, an increase in mitochondrial H_2_O_2_ levels in YAC128 mice and defects in Ca^2+^ handling [104] support an early increase in the striatal susceptibility of mitochondria. Furthermore, functional assays revealed an impairment of the mitochondrial respiratory capacity in the striatum and cortex of R6/1 mice. It is important to note that *N*-acetylcysteine administration delayed the onset and progression of motor deficits in R6/1 mice [208], which may have reduced both excitotoxicity and oxidative stress in the striatum. A recent study established that voluntary wheel running reduces hindlimb clasping in the R6/1 mouse model. Furthermore, chronic exercise in the R6/1 showed the clasping phenotype with normal mitochondrial respiration in the cortex and striatum, suggesting that mitochondrial dysfunction may not be necessary for the progression of symptoms [105]. At the molecular level, different studies explored the role of the mitochondria in apoptotic cell death processes with elevated caspase activity in HD mice models [106,192]. Interestingly, HTT has been shown to be involved in vitro with caspases-1, 2, and 3 [109,209,210]. In fact, immunostaining analysis of YAC72 mice displays an increase of caspase-2 in the SPNs within the striatum with concomitantly decreased levels of a brain-derived neurotrophic factor in the cortex and striatum at 3 months [209]. This indicates that caspase-2 participates selectively in the neurodegeneration of SNPs in the striatum. In contrast, in YAC128 mice models, the lack of caspase-2 results in protection from the well-validated motor and cognitive features of HD [211]. Moreover, previous studies in transgenic mice expressing exon 1 of the human HTT demonstrate that intracerebroventricular administration of a caspase inhibitor 1 delays disease progression [211] (Table 1).

In conclusion, while HD animal models have not yet provided definitive evidence whether or not mitochondrial bioenergy is critically involved in HTT-induced disease symptoms compared to PD animal models’ genes, mitochondrial bioenergy defects are nonetheless a major component of both diseases progression. On the other hand, the creation of HD mouse models using different strain backgrounds and expressing only a portion of the HTT protein leads to increased difficulty in understanding the compelling findings that were obtained with the genetic models of PD.

## 5. Mitochondrial Bioenergy in Parkinson’s Disease and Huntington Disease, Based on Human Evidences

Assessment of mitochondrial activity (or dysfunction) in the CNS of living patients is not easy. However, analyzing peripheral tissues and fluids can offer a reliable measure. Although the core pathology in PD and HD is central, a number of substantial abnormalities occur at the systemic or peripheral level. Peripheral blood mononucleate cells (PBMCs) are one of the most reliable “in vivo PD models”. Peripheral blood mononucleate cells present several alterations in critical metabolic pathways and accumulate α- synuclein in pathological forms, well recapitulating PD-related neuropathology [212,213]. In a recent study, the mitochondrial bioenergetics of PBMCs (Figure 2) from PD patients was assessed using the Seahorse Bioscience technology. The results showed a peculiar pattern of mitochondrial respiration, including normal basal respiration, significant augmentation of the maximal and spare respiratory capacities, and a tendency to increase ATP production. The increased spare respiratory capacity was found to follow the disease duration and severity of motor disturbances, revealing some mitochondrial adaptations to the higher bioenergetic requirements occurring at later disease stages [110]. Similarly, consistent findings of greater mitochondrial respiration were also observed in different cell lines obtained from PD patients, such as lymphoblasts (immortalized blood lymphocyte-derived cells) [214] and fibroblasts [215,216]. Therefore, it is reasonable to assume that in PD patients, unlike animal models, mitochondrial activity may increase in a compensatory manner. Indeed, the “nuclear factor erythroid 2-related factor 2” (Nrf2) pathway, a master regulator of cellular defense and mitochondrial activity, can be overexpressed in PD patients’ PBMCs in proportion to the disease duration, suggesting a systemic defensive response to the PD clinical-pathological progression [213]. Otherwise, changes in respiratory activity could reflect defects in mitochondrial structures, metabolism, or respiratory protein functions [217]. In another study, PBMCs showed increased glycolysis and deficits in superoxide dismutase, together with a peculiar mitochondrial vulnerability in the monocyte subpopulation in PD patients’ [218]. Substantial evidence can also be obtained from the assay of mitochondrial dysfunction biomarkers in PD patients’ fluids. Indeed, metabolomic studies performed in blood and cerebrospinal fluid (CSF) disclosed various abnormalities in metabolic pathways related to the mitochondria (in alanine, branched-chain amino acids, fatty acids, and acylcarnitines levels; in steroidogenesis; in the glutathione cycle) [219,220,221]. Likewise, the redox balance-related biomarker levels, such as uric acid [222,223,224,225], catalase, total glutathione [226], nonmercaptalbumin (an oxidized form of albumin) [227], oxidized glutathione [213], oxidized DJ-1 [228], α-klotho [229], lactoperoxidase [230], and heme-oxygenase-1 [231], were altered in the same fluids. Genomic studies performed on peripheral blood showed the down-regulation of genes critical for mitochondrial functions (COX4I1, ATP5A1, and VDAC3) [232], while the analysis of blood-circulating extracellular vesicles demonstrated the reduction in mitochondrial components (i.e., ATP5A, NDUFS3, and SDHB) in PD patients [233]. Furthermore, analysis of mitochondrial bioenergetics has also been performed in peripheral tissues of HD patients, with some controversial findings depending on the matrix or the experimental technique used. Indeed, mutant huntingtin is expressed outside the CNS as well, accounting for the substantial impairment in multiple cellular pathways related to the mitochondria and redox balance [234] (Figure 2). It has also been observed that skin fibroblasts may show some energetic, respiratory, redox, and morphological abnormalities [235,236]. Likewise, several functional or structural mitochondrial defects have been noticed in HD patients lymphoblasts [237]. Finally, increased oxidative damage, reduced antioxidant capacity, and mitochondrial abnormalities have been tracked even in HD patients’ blood through different fluid markers [238].

## 6. Discussion

Degenerative diseases commonly entail mitochondrial impairment. The mitochondria represent the major source of bioenergy for the cell, and as a consequence, their putative failure leads to a lack of energy. Daily energy metabolism for both humans and organisms is the sum of daily energy expenditure and daily energy intake. Daily energy expenditure may be divided into different categories: (1) energy spent on basal metabolism; (2) energy spent by thermic increment following food intake; (3) energy spent on thermoregulation; and (4) energy spent during daily activities [239]. In particular, energy metabolism is the process of generating energy (ATP) from nutrients, and mitochondria represent the main powerhouse of the cell. To produce energy in the form of ATP and GTP, all the cells, especially the neurons, consume glucose, amino acids, and fatty acids [239]. These nutrients are processed and transferred into the tricarboxylic acid (TCA) cycle, and electrons are stored in their reducing equivalents, NADH and FADH2, through iterative oxidations. NADH and FADH2 serve as carrier molecules that transport electrons into the electron transport chain (ETC), while protons flow down to generate ATP [240]. The capacity of the mitochondria to supply energy in the form of ATP when required could be referred to as bioenergetic efficiency, which may change due to the effects of environmental stress, aging, or pathologies. For example, during aging, the mitochondria undergo alterations in their capacity to produce ATP [240,241]. This is due to the release of a large number of reactive oxygen species (ROS). ROS are able to increase spontaneous DNA mutations and can start the processes that lead to cancer [241,242].

Mitochondrial biogenesis is a process that continuously balances the degradation of dysfunctional mitochondria inside the cells. The disruption of this homeostasis, which may be guaranteed in all functioning cells, represents an additional point that focalizes the loss of energetic function, which may precede cellular loss. This critical balance is called mitochondrial dynamics. During these processes, mitochondrial fusion and fission regulate the number and size of the mitochondria, while mitochondrial biogenesis increases the mitochondrial mass inside the cells by producing ATP as a response to greater energy demand. Mitochondrial biogenesis requires a very multifaceted process that starts with the coordination of the nuclear and mitochondrial expression programs. The regulator for the cited process is the protein PGC-1α, which is able to interact with many transcription factors, such as myocyte enhancer factor 2C (MEF2C), nuclear respiratory factors 1 and 2 (NRF1 and NRF2), peroxisome proliferator-activated receptors (PPAR), estrogen-related receptor alpha (ERRα) and many others involved in coordinating mitochondrial biogenesis and oxidative metabolism [243,244,245]. Moreover, NRF1 and NRF2 cooperate to upregulate the transcription of several nuclear-encoded genes that act in the negative feedback effect on mitochondrial production [244,246]. Selective autophagic degradation, called mitophagy, recruits the damaged mitochondria into a pre-autophagosome structure via a PINK1/Parkin-dependent process. Indeed, the mitochondrial transport system is essential for the distribution of the mitochondria into the cells and, in particular, into the different neuronal structures, which require the synthesis of the mitochondria and their proteins [247,248,249,250].

A consistent number of mitochondria are localized at the presynaptic terminals, to provide ATP for intense synaptic activity [251]. In fact, it was hypothesized that the loss of synaptic plasticity found in PINK1 KO mice was a result of the lack of ATP at the presynaptic terminal, thereby confirming the essential role of mitochondria in this area [252].

Previously, we have referred to the role of the PINK1 gene in early inherited parkinsonism, and these findings are in support of the theory that links mitochondrial dysfunction to PD.

In addition, the mitochondria in aging people show different features [253]; they swell while their numbers dwindle and are unable to replace themselves as quickly in their dysfunctional state [254]. Parkinson’s disease is also an aging disease, in its idiopathic form, and this suggests a link with the common non-genetic form of PD. Therefore, failure in the mitochondria is the starting phase of the entire neurodegenerative process [255]. 

It was demonstrated in both PD and HD that the presence of environmental stress can increase the production of oxidative species and can also modify the structure of proteins and DNA inside the mitochondria and its nuclei [256]. Usually, HD is considered a hyperkinetic disorder despite the presence of hypokinetic features in the motor symptoms [250]. Conversely, PD is considered a hypokinetic disorder, in which the resultant speech motion disease may be classified as hyperkinetic or hypokinetic dysarthria [257,258]. Both are defined as neurodegenerative disorders causing the gradual loss of neurons, and we can explain why. Concerning HD, the mutant HTT protein was shown to be able to directly impair the ability of PGC-1α to activate target genes related to mitochondrial biogenesis and normal mitochondrial function. The amplification of mHTT proteins enhances mitochondrial depolarization and swelling, which are calcium dependent. In addition, it has been demonstrated that mutant HTT’s ability to combine specifically with the beta-tubulin subunit can adversely affect mitochondrial transport. Moreover, both disorders result in a deficit at the level of the basal ganglia, with peculiar references to the striatum [259] and hippocampal areas [260]. 

In rodent models of HD, only the progression of the disease leads to neuronal loss. Cell dysfunctions may generate symptoms, and in particular, abnormal synaptic statements due to a lack of ATP [261]. In contrast, in human patients, clinical symptoms occur as a combination of functional and structural changes that may result in a reduction in lactate levels in the brain area as a consequence of an altered mitochondrial metabolism [250].

In PD models, dysfunctions in striatal plasticity are described in detail [135] and are strictly connected to energy failure [132,135].

These areas are linked to movement and cognitive ability. For this reason, it is not a surprise that both diseases can affect cognitive or thinking abilities and motor functions. Studies on rodent models of PD show impairment in the bioenergetic pathway and a block in function in mitochondrial complexes I and III. Additionally in genetics, phenotypic models were induced by toxin exposure in the early stages of the disease as well as in the developmental stage [86,94,122,124,125,238]. On the other hand, studies on animal models of HD show an evident alteration in mitochondrial complexes II and III and describe a reduction in bioenergetic efficiency that was described essentially in the early pathogenic mechanism [262]. 

Interestingly, this evidence has been confirmed by translational studies in men, in which the association between mitochondrial deficit and biomarkers of PD and HD was shown [252]. Actually, there is still no cure for pathology, and drugs that hinder the development and severity of the diseases do not exist. Consequently, the knowledge of pathogenetic mechanisms and the similarities between these diseases can help us while reviewing the traditional approach in order to develop novel approaches. In fact, new pharmacological approaches focused on mitochondrial target agents could be used in the future to cure degenerative and, in particular, neurodegenerative pathologies.

## 7. Future Perspectives

Numerous evidences show the role of mitochondria in neurodegenerative diseases. Approximately 1209 papers described the mitochondria’s involvement in non-neurological degenerative processes (cancer, aging, and other pathologies) (Figure 2A,B). Among the published works on PD, 53% (of 5185 papers) talked about mitochondrial involvement in the pathogenesis of the disease, while only 5% focused on the bioenergetic role of the mitochondria and the regulatory and protective role of the PINK1 gene in the mitochondrial machinery (Figure 2C). In HD, on the other hand, 79% (of 1031 works) indicated the involvement of the mitochondria in the progression of symptoms and only 1% highlighted the protective role of the PINK1 gene in the mitochondrial machinery (Figure 2D).

Conversely, many questions remain unresolved, especially about how mitochondrial failures progress and the loss of energy efficiency triggered by environmental events. However, it is clear that in these two illnesses, the role of mitochondria in providing the energetic functions of the neuron is still lacking [4]. 

Therefore, in-depth studies in this field could provide more insight and knowledge about movement disorders and could offer new therapeutic solutions for pathologies, such as PD and HD, for which only symptomatic cures currently exist.

## Figures and Tables

**Figure 1 ijms-24-07221-f001:**
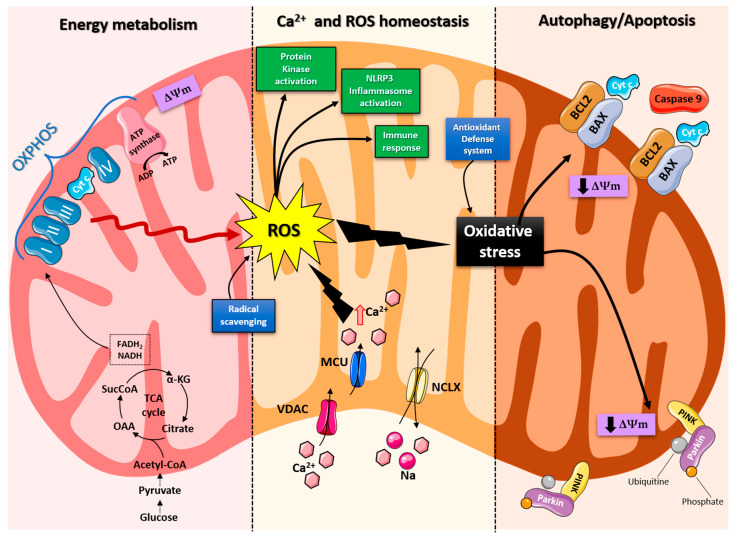
Schematic representation of the mitochondrial functions. From left: Under physiological conditions, mitochondria supply ATP through OXPHOS. Krebs cycle enzymes use acetyl-coenzyme A to reduce NADH and FADH2, which are used for energy transfer to the electron transport chain (ETC) embedded in the inner mitochondrial membrane. OXPHOS is also an important source of ROS, whose basal levels are maintained by the radical scavenging network. Mitochondria also play a crucial role in calcium homeostasis. The voltage-dependent anion channel (VDAC) and the mitochondrial Ca^2+^ uniporter complex (MCU) finely control Ca^2+^ passage across the mitochondrial membranes, while the mitochondrial Na^+^/Ca^2+^ exchanger (NCLX) is one of the central units involved in Ca^2+^ extrusion. Under normal conditions, through ROS generation and redox signaling, mitochondria can control cellular metabolism, physiology, the inflammatory response, and immune function and act as important signaling molecules in the cell by activating various protein kinases. In contrast, the overproduction of ROS and dysregulation of the redox signaling system result in oxidative stress that can lead to mitochondrial damage. Malfunctioning mitochondria can be selectively removed through mitophagy, or, as all other defense mechanisms fail, the neuron can orchestrate its own destruction by activating the intrinsic suicide program of apoptosis.

**Figure 2 ijms-24-07221-f002:**
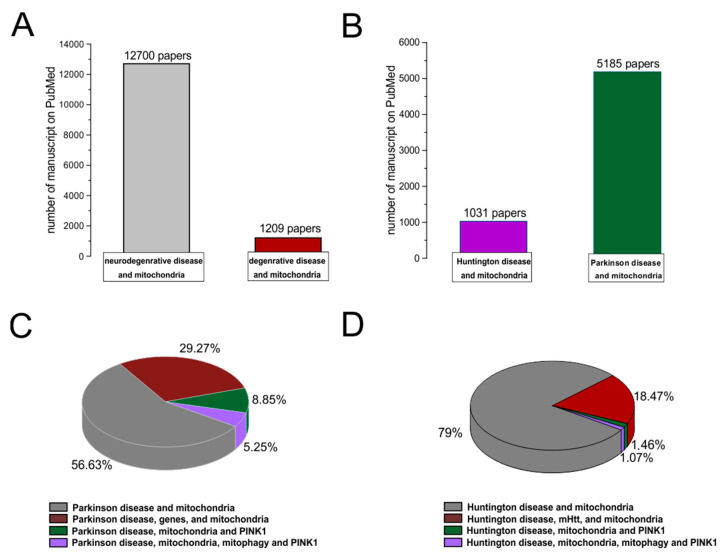
PubMed evidences of mitochondrial involvement in neuro-degenerative/degenerative diseases. Records were obtained from research articles published until January 2023. Article publications were obtained from the following PubMed searches: “Neurodegenerative disease and mitochondria; degenerative disease and mitochondria (**A**); HD and mitochondria; PD and mitochondria (**B**)”. (**C**,**D**) To realize the pies on sub-items, we have selected “PD, genes, and mitochondria; PD, PINK1, and mitochondria; PD, PINK1, mitophagy, and mitochondria; HD, mHHT, and mitochondria; HD, mHHT, mitophagy, and mitochondria.

**Figure 3 ijms-24-07221-f003:**
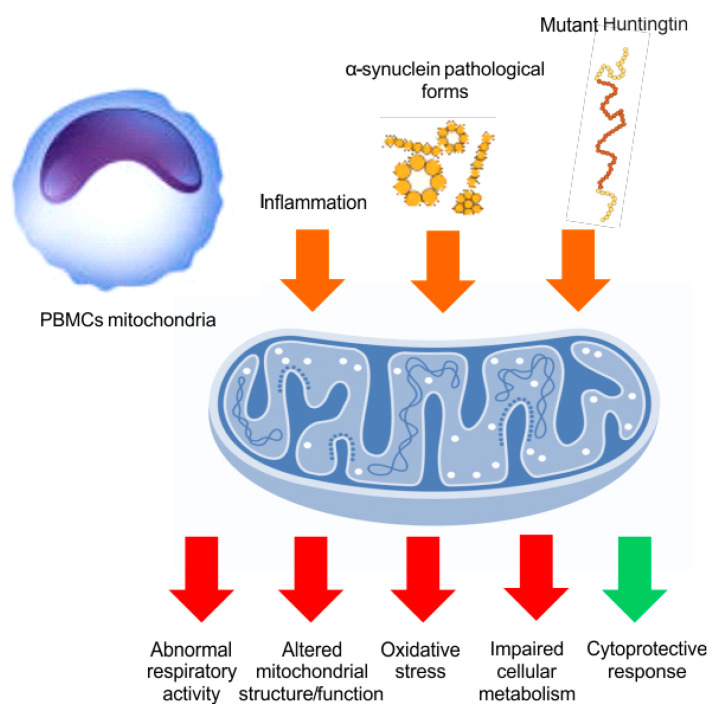
PBMCs mitochondrial function. In PD and HD, the systemic inflammation and the circulating neurodegeneration-related peptides (orange arrows) might affect the PBMCs mitochondrial function, inducing altered function or morphology, abnormal respiratory activity, increased oxidative stress, and impaired cellular metabolism (red arrows). However, some protective mechanisms could also be triggered (green arrow), especially at early disease stages, although they would be later overwhelmed by clinical-pathological progression.

**Table 1 ijms-24-07221-t001:** Alteration of mitochondrial bioenergy function identified in genetic mouse models of PD and HD. The table summarizes the major mitochondrial changes found in murine models of PD and HD. The legend of the abbreviations is listed below.

	Mouse Model	Mitochondria Alteration	Molecules	References
PD	α-synuclein A53T Mouse (Tg)	EN-mt	↓ Drp1, ↓ Mfn1	[90]
		mtDNA damage	c-caspase-3 and p53	[91]
			↑ Mfn1, ↓ Mfn2	[92]
	A53T-hα-syn	mtUA	↑ PRKAG2, ↑ TTR	[93]
		DE-autophagic/endocytic DA fibres		[93]
		altered TCA cycle at striatal synapses		[93]
	PINK1 KO mouse	↑ number of larger mt		[94]
		↓ respiratory complex I, II, III activity, age dependent; ↓ CAA and TCA cycle activity; ↑ protein oxidation	_	[94]
	PINK1 KO rat	↓ ATP production	↑DRP1	[95]
		defects complex I	↑ O_2_ consumption	[95]
		increased complex II		[95]
			bioinformatic analysis, PGC1A, PG1B, TFAM, GF1R, INSR, pathways were deactivated	[95]
	DJ-1 mouse KO		↓ aconitase	[96,97,98,99]
			activity; ↑ROS	
			production	
			↑ Ca	[96,97,98,99]
			↑ GSH level and ↑ GSH/glutamate ↑ Glu	[96,97,98,99]
			↑ TCA cycle, H_2_O_2_ consumption ↑ mitochondrial Trx activity, ↑ GSH and ↑ GSSG, ↑ GRX ↓ GR	[96,97,98,99]
	Parkin mouse KO	DP, Cell Stress Chaperones and UPP components		[100]
		↓ subunits of complexes I ↓subunits IV	↓ peroxide reductases	[101]
			↓ antioxidant capacity ↓ protein of lipid peroxidation	[101]
HD	R6/1 mouse	↑ (ΔΨm)	↑ Ca^2+^, ↑ NAD(P)H	[102]
	R6/2 mouse		↑ OH(8)dG	[103]
			↓ in NAA	[104]
			↑ glutamine ↑ glucose	[105]
			↑ creatine	[106]
			↑ GPC, ↑ glutamine and ↑ glutathione ↓ AA decreased at 8 weeks	[106]
		reduction in mt complex IV activities (12 weeks)	↑ iNOS and ↑nitrotyrosine	[107]
			↓ aconitase cerebral cortex	
		↓ decrease in mitochondrial mass	synaptosomal ↑ ROS production and ↑ antioxidant in striatum	[108]
	YAC128 mouse	↑ basal and maximal mitochondrial respiration	↑ [64Cu]-ATSM	[109]
			↑ ATP production, and ↑ complex II and III	[109]
			↑ oxygen consumption rate	[109]
			↓ Ca handling	[109]
	YAC72 mouse		↑ caspase-2	[110]

Parkinson disease (PD); Huntington disease (HD); decrease ↓; increase ↑; Mitochondrial membrane potential (ΔΨm); Mitochondria: mt; Transgenic animal model of human α-synuclein: α-synuclein A53T Mouse (Tg); AAV-mediated overexpression of α-synuclein: A53T-hα-syn; Enlarged neuronal mitochondria: EN-mt; Mitochondrial DNA damage: mtDNA damage; cleaved caspase-3: c-caspase-3; mitochondrial ultrastructural abnormalities: mtUA; disturbances exhibit autophagic: DE-autophagic; endocytic dopaminergic fibres: endocytic DA fibres: mitochondrial respiration or Krebs cycle: TCA cycle; cytosolic aconitase activity: CAA; proteins involved in detoxification: DP; stress-related chaperones: Cell Stress Chaperones; ubiquitin-proteasome pathway: UPP; 8-hydroxy-2-deoxyguanosine: OH(8)dG; *N*-acetylaspartate: NAA; Acetylaspartate: AA; Glycerophosphorylcholine: GPC; Inducible nitric oxide synthase: iNOS; aconitase cerebral cortex: Acon-CC; oxygen consumption rate: OCR.

## Data Availability

The data presented in this study are available in PubMed library.

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
