# Peer review of "Mitochondrial Bioenergy in Neurodegenerative Disease: Huntington and Parkinson"

_ijms, 2023, doi:10.3390/ijms24087221_

Round 1

Reviewer 1 Report (Previous Reviewer 3)

The manuscript has been revised well.

Author Response

RE: We are very grateful to the referee for her/his help during the peer review process. His/Her suggestion was very precious.

Reviewer 2 Report (Previous Reviewer 4)

Dear authors,

thank you for all corrections.

Please correct the line 12 in the Abstract.

Please correct all  in-text citations according to IJMS.

Author Response

Dear authors,

thank you for all corrections.

Please correct the line 12 in the Abstract.

RE: We want to thank the referee for her/his suggestion. The correction was done.

Please correct all  in-text citations according to IJMS.

RE: We thank the referee for his/her valuable suggestion. In the present version of Manuscript text citation were correct according to IJMS

All the changes are underlined in green.

This manuscript is a resubmission of an earlier submission. The following is a list of the peer review reports and author responses from that submission.

Round 1

Reviewer 1 Report

The authors need to improve the language of the manuscript from the Abstract to the conclusion. 

Author Response

Reviewer 1

Comments and Suggestions for Authors

The authors need to improve the language of the manuscript from the Abstract to the conclusion. 

RE: We are very grateful to the referee for her/his suggestion and asked a native speaker, an expert in the field, to revise the manuscript. We hope that this version will be more comprehensible and easy to read.

Reviewer 2 Report

Authors purpose a review in which they emphasize how mitochondrial dysfunctions are associated with phenomena as inflammation and degeneration, common elements of the two major neurodegenerative diseases, Huntington’s disease and Parkinson’s disease. Therefore, new pharmacological approaches considering mitochondria target agents could be used in the future to cure degenerative and in particular neurodegenerative disorders with common elements.

The manuscript needs to be better organized using subheading and organizational paragraphs to lead the reader through the different topics.

Authors should include a figure which summarize the role of mitochondria in brain energy metabolism, calcium homeostasis, and signal transduction.

The meaning of Figure 1 is unclear. Authors should improve the legend to figure; moreover, they should specify the meaning of the colored arrows. Which kind of software was used to make the figure?

Authors cite “table 1” several times, but it is missing in the manuscript.

Line 47: replace “highenergy” with “high-energy”.

Line 55: replace “indifferent” with “in different”.

Line 353: replace “C+” with “Ca2+”.

Author Response

Reviewer 2

Comments and Suggestions for Authors

The authors purpose a review in which they emphasize how mitochondrial dysfunctions are associated with phenomena such as inflammation and degeneration, common elements of the two major neurodegenerative diseases, Huntington’s disease and Parkinson’s disease. Therefore, new pharmacological approaches considering mitochondria target agents could be used in the future to cure degenerative and in particular neurodegenerative disorders with common elements.

The manuscript needs to be better organized using subheading and organizational paragraphs to lead the reader through the different topics.

RE: We want to thank the referee for her/his suggestion. In the present version of manuscript the subheading were added. We hope now the paper will be easier to understand.

Authors should include a figure which summarize the role of mitochondria in brain energy metabolism, calcium homeostasis, and signal transduction.

RE: We agree with the suggestion of the referee, and we took the opportunity to include in the text three new figures, as well as the table that was lost.

The meaning of Figure 1 is unclear  Authors should improve the legend of to figure; moreover, they should specify the meaning of the colored arrows. Which kind of software was used to make the figure?

RE: We thank the referee for his/her valuable suggestion. In the present version of the Manuscript, figure 1 was been removed and substituted with new figures and legends (placed on the last pages of the manuscript).

The authors cite “table 1” several times, but it is missing in the manuscript.

RE: We thank the referee for his/her suggestion; the table is now uploaded into the text.

Line 47: replace “highenergy” with “high-energy”.

RE: We agree with the referee suggestion. We have substituted the words in according with the request.

Line 55: replace “indifferent” with “in different”.

RE: We thank the referee. We have substituted the words according to the request.

Line 353: replace “C+” with “Ca2+”.

RE: We agree with the referee's suggestion. We have substituted the words according to the request.

Reviewer 3 Report

In this review, the authors summarize the characteristics of mitochondrial dysfunction shared between Huntington's disease and Parkinson's disease. Although much of the literature has been read and much effort paid, there are still some questions to be considered.

1. Please add the “Abbreviations” in the manuscript.

2. Please consider adding some charts about mitochondrial function and action.

3. It is suggested to sort out the structure of the manuscript and change the long paragraphs to short paragraphs in order to reduce reading fatigue.

4. The discussion is very important for a high-quality review. Please strengthen this in the conclusion.

5. It is better to add several figures or tables about the contents in the manuscript in order to provide the information more clearly.

Author Response

Reviewer 3

Comments and Suggestions for Authors

In this review, the authors summarize the characteristics of mitochondrial dysfunction shared between Huntington's disease and Parkinson's disease. Although much of the literature has been read and much effort paid, there are still some questions to be considered.

  1. Please add the “Abbreviations” in the manuscript.

RE: We are very grateful to the referee for her/his suggestion; we have added a list of abbreviations.

  1. Please consider adding some charts about mitochondrial function and action.

Re: We agree with the suggestion of the referee, and we have added 3 new figures.

  1. It is suggested to sort out the structure of the manuscript and change the long paragraphs to short paragraphs in order to reduce reading fatigue.

RE: We want to thank the referee for her/his suggestion. In the present version of the manuscript, the subheading was added. We hope now the paper will be easier to understand.

  1. The discussion is very important for a high-quality review. Please strengthen this in the conclusion.

RE: We agree with the suggestion of the referee; the conclusions have been strengthened to highlight how mitochondrial bioenergetics is the headmaster of degenerative diseases.

  1. It is better to add several figures or tables about the contents of the manuscript in order to provide the information more clearly.

RE: We agree with the referee, and in the present version of the manuscript we have added 3 new figures and 1 table.

Reviewer 4 Report

In the review the authors discussed the theory that both Parkinson’s disease and Huntington’s disease are start and develop in light of mitochondrial dysfunction.

This manuscript is principal interesting, but need some important changes before publishing may be possible.

General points:

This manuscript needs correction by a native speaker.  

Please add a list of abbreviations before References section to your manuscript.

Please add a Future perspectives section to your manuscript.

Special points:

Please change the title of your manuscript to: Mitochondrial bioenergy in neurodegenerative disease: Huntington and Parkinson

Please add 2 more Figures to your manuscript.

Why some references numbers in your manuscript are blue and some are black? Please correct. Please also correct all spaces between the words and references numbers in the whole manuscript.

Keywords: please also add to keywords: Parkinson’s disease; Huntington’s disease.

Introduction

Lines 31-62: please add multiple references at the end of each these sentences.

Main text of the manuscript

Line 200: please change the title to: Role of mitochondria in neurodegenerative disease

Lines 292-299: please add multiple references at the end of each these sentences.

Lines 301-307: please describe all these studies exactly.

Lines 538-548: please add multiple references at the end of each these sentences.

Discussion

Lines 588-634: please add multiple references at the end of each these sentences.

References

Please do your References list according to IJMS.

Author Response

Reviewer 4

Comments and Suggestions for Authors

In the review, the authors discussed the theory that both Parkinson’s disease and Huntington’s disease start and develop in light of mitochondrial dysfunction.

 This manuscript is principally interesting but needs some important changes before publishing may be possible.

 General points:

 This manuscript needs correction by a native speaker.  

RE: We are very grateful to the referee for her/his suggestion and asked a native speaker, an expert in the field, to revise the manuscript. We hope that this version will be more comprehensible and easy to read.

Please add a list of abbreviations before the References section of your manuscript.

RE: We want to thank the referee for her/his suggestion. A list of abbreviations was included before References.

 Please add a Future perspectives section to your manuscript.

RE: We are very grateful to the referee for her/his suggestion. In the present version of manuscript, we have added a new paragraph called “future perspective”.

Special points:

 Please change the title of your manuscript to Mitochondrial bioenergy in neurodegenerative disease: Huntington and Parkinson

RE: According to the referee, we have changed the Title. We are grateful for the suggestion.

 Please add 2 more Figures to your manuscript.

RE: We want to thank the referee for her/his suggestion. In the present version of the manuscript old figure, one was being, removed and 3 new figures are added.

 Why some references numbers in your manuscript are blue and some are black? Please correct. Please also correct all spaces between the words and references numbers in the whole manuscript.

RE: We apologize for the mistake; now all references are in black.

 Keywords: please also add to keywords: Parkinson’s disease; Huntington’s disease.

RE: According to referee's request the keyword is added.

Introduction

Lines 31-62: please add multiple references at the end of each of these sentences.

RE: We want to thank the referee for her/his suggestion. The references were added.

 The main text of the manuscript

Line 200: please change the title to Role of mitochondria in neurodegenerative disease

RE: We want to thank the referee for her/his suggestion. The paragraph was renamed accordingly.

Lines 292-299: please add multiple references at the end of each of these sentences.

RE: We thank the referee for her/his suggestion. The references were added in the present version of the manuscript.

Lines 301-307: please describe all these studies exactly.

RE: According to the referee, we have reorganized the phrase exactly.

Lines 538-548: please add multiple references at the end of each of these sentences.

RE: We want to thank the referee for her/his suggestion. The references were added.

 Discussion

Lines 588-634: please add multiple references at the end of each of these sentences.

RE: According to the referee's suggestion, we have added new references.

 References

Please do your References list according to IJMS.

RE: We thank the referee for her/his suggestion, we have revised the references list according to IJMS dispositions.

Round 2

Reviewer 2 Report

The revision has significantly improved the manuscript, which is now suitable for publication.

Reviewer 4 Report

Dear authors, thank you for all corrections.

I have another two important points:

1. Lines 32-43: please really add the multiple references at the end of each sentences.

2. Table 1 has a very bad quality. Please correct and improve the quality of the Table 1.

Please add a apopriate abbreviations below the Table 1.